# Autoantibody Biomarkers in Rheumatic Diseases

**DOI:** 10.3390/ijms21041382

**Published:** 2020-02-18

**Authors:** Eun Ha Kang, You-Jung Ha, Yun Jong Lee

**Affiliations:** 1Division of Rheumatology Department of Internal Medicine, Seoul National University Bundang Hospital, Seongnam 13620, Korea; hayouya@snubh.org (Y.-J.H.); yn35@snu.ac.kr (Y.J.L.); 2Department of Internal Medicine, Seoul National University, Seoul 03080, Korea

**Keywords:** autoantibody, biomarker, rheumatic disease

## Abstract

Autoantibodies encountered in patients with systemic rheumatic diseases bear clinical significance as a biomarker to help or predict diagnosis, clinical phenotypes, prognosis, and treatment decision-making. Furthermore, evidence has accumulated regarding the active involvement of disease-specific or disease-associated autoantibodies in the pathogenic process beyond simple association with the disease, and such knowledge has become essential for us to better understand the clinical value of autoantibodies as a biomarker. This review will focus on the current update on the autoantibodies of four rheumatic diseases (rheumatoid arthritis, myositis, systemic sclerosis, and anti-neutrophil cytoplasmic antibody associated vasculitis) where there has been a tremendous progress in our understanding on their biological effects and clinical use.

## 1. Introduction

Biomarkers are not only helpful for understanding distinct pathophysiological processes of the disease but also useful for patient care in terms of screening and early diagnosis of the disease, prediction of prognosis, and treatment decision-making. Autoantibodies in systemic rheumatic diseases have been biomarkers largely due to significant and sometimes exclusive association with the disease itself or certain clinical phenotypes of the disease. In addition, recent studies have shown active involvement of these autoantibodies in the disease process beyond simple association. This review will discuss updated information on the pathogenic roles of autoantibodies in four systemic rheumatic diseases: rheumatoid arthritis (RA), myositis, systemic sclerosis (SSc), and anti-neutrophil cytoplasmic antibody associated vasculitis (AAV) and their values to guide diagnosis, prognosis, and treatment of corresponding diseases.

## 2. Autoantibodies in Rheumatoid Arthritis

RA is an autoimmune-mediated chronic systemic inflammatory disease that primarily affects synovial joints but often shows extraarticular manifestations. Autoantigens targeted by autoantibodies found in RA display a wide spectrum of cellular components, suggesting that RA is characterized by accumulated autoreactivities in both B and T cells [1]. Among diverse autoantibodies found in patients with RA, rheumatoid factor (RF) and anti-citrullinated protein antibody (ACPA) are the two most remarkable autoantibodies in RA that provide useful clinical and pathophysiological information. Currently, they are included in the 2010 American College of Rheumatology/European League Against Rheumatism (ACR/EULAR) classification criteria for RA [2]. 

### 2.1. Pathogenic Roles of RFs 

RFs, directed to the Fc portion of IgG, are normally induced by immune complexes (ICs) and polyclonal B cell activators such as bacterial lipopolysaccharides and the Epstein-Barr virus [3]. The physiological roles of RFs are: (1) to enhance IC clearance by increasing its avidity and size, (2) to help B cells uptake IC, and thereby, efficiently present antigens to T cells, and (3) to facilitate complement fixation by binding to IgG-containing ICs [4]. In RA, RFs are locally produced by B cells in lymphoid follicles and germinal center-like structures from the inflamed synovium [5,6]. RFs are detected in 60%–80% of RA patients [7]. RA-associated RFs are characterized by affinity maturation [8], with increasing diagnostic specificity at higher titers (e.g., IgM RF ≥ 50 IU/mL) [7] and by IgA isotypes [7,9]. High titer RFs are also indicators for radiologic erosion [7], which indicates a role of RFs in RA pathogenesis. The capacity of RFs to enhance IC formation with preexisting autoantibodies, including ACPAs, may potentiate the arthritogenicities of these autoantibodies and perpetuate inflammation [10,11,12,13]. In line with this idea, ACPA-containing IC (ACPA-IC) induced greater proinflammatory responses from macrophages when they were formed in the presence of IgM or IgA RFs [10,11]. Similarly, the co-presence of RFs and ACPAs showed an additive effect on the erosion burden among RA patients [12], as well as higher disease activity [11]. Consequently, there could be a vicious cycle among ACPA generation, immune complex formation, RF production, and chronic inflammation, eventually leading to tissue damage.

### 2.2. Pathogenic Roles of ACPAs

ACPAs are a group of autoantibodies found in 70%–90% of RA patients [4]. They are directed towards citrullinated proteins generated by post-translational modification on arginine residues and, therefore, can recognize a variety of articular and non-articular antigens, including type II collagen, fibrinogen, vimentin, and others [4]. Importantly, citrullination results from peptidyl-arginine deiminase (PAD) induced by inflammation [4]. Compared to RFs, ACPAs have a higher disease specificity (70%–80% versus 90%–95%) [4]. They are primarily generated among genetically susceptible individuals or shared epitope (HLA-DR4) carriers as a result of environmental stimuli, such as smoking, that induce a series of events: chronic inflammation, PAD induction, citrullinated autoantigen expression, and high-affinity loading of citrullinated proteins on shared epitope molecules of antigen-presenting cells [14]. Similar to RFs, they are associated with the radiologic progression of RA [7]. The pathogenic role of ACPAs has become more evident by an animal model where transferred antibodies directed to citrullinated fibrinogen or citrullinated collagen led to disease exacerbation in mice with mild experimental arthritis [15,16]. According to the current knowledge, the pathogenic roles of ACPAs can be summarized as follows.

#### 2.2.1. Fc Receptor Binding 

IgG antibodies exert an effect via multimeric interactions with the Fc receptor (FcR) by forming ICs [17]. Likewise, evidence suggests that ACPAs contribute to RA inflammation via IC formation with citrullinated proteins [18,19,20]. For example, citrullinated fibrinogen-containing ICs induce tumor necrosis factor α (TNFα) secretion from macrophages in FcγR and Toll-like receptor 4 (TLR4)-dependent manners, where both protein citrullination (compared to native protein) and IC formation greatly potentiates TNFα secretion from macrophages [18]. These results support the hypothesis that ICs containing citrullinated peptides may serve as danger-associated molecular patterns (DAMPs) to stimulate innate immune cells. Considering that RA autoreactivity also targets other potential DAMPs of endogenous origin (e.g., heat shock proteins) [21], the roles of other pattern recognition receptors than TLR4 and the differences between citrullinated and native DAMPs than fibrinogen need future research.

In addition to IC formation with citrullinated proteins and induction of cytokine secretion, FcR engagement by ACPAs are also able to polarize macrophages to the pro-inflammatory M1 subset through interferon regulatory factors 4 and 5 pathways, leading to a higher M1/M2 ratio in RA patients than in osteoarthritis patients [22].

#### 2.2.2. Complement Activation

The key functions of the complement system include clearance of microorganisms and circulating ICs through opsonization [23]. Three pathways exist by which the complement system is activated, each utilizing different recognition molecules to initiate activation: classical, alternative, and lectin pathways [23]. All of these pathways can be activated by ICs, implying that complement activation could be an important part in autoantibody-associated rheumatic diseases, such as RA. In a serum transfer mouse model for RA, anti-glucose-6-phosphate isomerase IgG alone could induce arthritis through FcR and the complement network in lymphocyte-deficient recipients, supporting the importance of the complement system in RA pathogenesis [24]. Complement activation products, such as C3a, C5a, or the membrane attack complex (MAC), were also shown to play an important role in collagen-induced arthritis [25]. 

Complement consumption is observed in the synovial fluid of RA patients [23]. Both RFs and ACPAs can independently induce complement activation [26,27], and ACPAs have the ability to activate both classical and alternative pathways [27]. Moreover, it has been shown that complement activation by ACPA-IC is more prominent when ACPA-ICs are formed in the presence of IgM or IgA RFs purified from RA sera [10]. 

#### 2.2.3. NET Formation

Neutrophil extracellular traps (NETs) are networks of extracellular fibers primarily composed of highly condensed chromatins and antimicrobial peptides released from neutrophils, the key function of which is to trap and kill pathogens [28]. NETosis is a form of programmed cell death characterized by release of NETs. Diverse triggers, including microbial and nonmicrobial (autoantibodies, immune complex, cytokines, or other stimuli) components can induce NET formation or NETosis via TLRs, FcRs, or complement receptors on neutrophils [29]. Importantly, NETs expose potential intracellular autoantigens, which are normally of limited access by antigen-presenting cells or circulating autoantibodies. Thus, particular attention has been placed on the role of NETs in inducing autoantibodies for cytoplasmic and nuclear antigens. According to the analysis by Khandpur et al., NETosis is another mechanism by which ACPAs or RFs contribute to RA pathogenesis [30]. Enhanced NETosis was observed from both circulating and synovial neutrophils among RA patients compared to healthy controls or patients with osteoarthritis. ACPAs and RFs, as well as key proinflammatory cytokines of RA, independently led to NETosis, and the degree of netting neutrophils was correlated with ACPA levels [30]. Since the PAD inhibitor inhibited cytokine-induced NETosis, the PAD pathway is also important in NETosis. In line with this, NETs were found to externalize citrullinated autoantigens (e.g., vimentins) [30]. Thus, ACPA and/or RFs may perpetuate RA inflammation via NET formation, which in turn induces ACPA responses by exposing citrullinated proteins. 

#### 2.2.4. Osteoclastogenesis

ACPAs can directly stimulate osteoclast and enhance osteoclastogenesis [31]. IgG ACPAs isolated from RA patients targeting different citrullinated peptides were found to enhance osteoclastogenesis through a PAD-dependent IL-8-mediated manner, indicating that the stimulation ability was related to citrullinated peptides recognized by IgG ACPAs [32]. Consequently, administration of ACPAs to mice led to osteopenia and increased osteoclastogenesis [31,32]. 

Unlike ACPAs, the evidence suggesting a direct effect of RFs on osteoclastogenesis is lacking. However, RFs may indirectly contribute to osteoclastogenesis in patients with RA via the facilitated release of proinflammatory cytokines (e.g., TNFα) from immune cells [10,11]. Moreover, activation of FcRs on osteoclast precursors by crosslinked antibodies were found to enhance osteoclast formation [33]. Thus, RFs may be involved in increased osteoclastogenesis in RA via FcR engagement as well. Indeed, clinical observation showed that the effects of ACPAs and RFs were additive on bone erosions of RA patients [12].

#### 2.2.5. Pain Induction

While pain may not be not a causal factor for RA, it often precedes clinical or ultrasonic synovitis. According to the study by Wigerblad et al., IL-8 released from osteoclasts in an autoantibody-dependent manner produces pain by activating sensory neurons independently of inflammation, which may explain the disconnect between pain and inflammation in certain settings of RA [34]. 

### 2.3. Assays to Detect ACPAs

ACPAs are usually measured by anti-cyclic citrullinated peptide (anti-CCP) assays. Based on the finding that the anti-perinuclear factor, an autoantibody highly specific for RA, is directed to citrullinated filaggrin, the first-generation anti-CCP (anti-CCP1) utilized filaggrin-derived cyclic rather than linear citrullinated peptides to improve antibody recognition but was not widely used due to low sensitivity [35]. Unlike anti-CCP1, the second and third generation tests (anti-CCP2 and anti-CCP3) are no longer based on filaggrin-derived peptides but on a mixture of synthetic cyclic peptides selected from libraries of citrullinated peptides that mimic conformational epitopes [36]. It is well-established that the anti-CCP2 possesses a superior sensitivity (around 70%–80%) than anti-CCP1 while maintaining very high specificity (98%–99%) [35]. Until now, anti-CCP2 has been considered as the gold standard of testing for ACPA [35]. In recent years, new ACPA tests, such as the anti-CCP3 assay, anti-mutated citrullinated vimentin, or others, have been commercially introduced. However, they have not shown a clear advantage over the anti-CCP2 assay [35]. 

### 2.4. RFs and ACPAs as Biomarkers

The most common use of RFs or ACPAs as biomarkers is to divide RA patients into seropositive (RF and/or ACPA positive) versus seronegative subsets that differ in risk factors, clinical outcomes, and treatment response. The risk factors of the seropositive RA and those of ACPA formation are in fact overlapped (e.g., genetic risk factors such as shared epitope and smoking) [4]. While both RF and ACPA associate with radiographic damage compared to the seronegative status, the former associates more with extra-articular manifestations than the latter [37]. As expected from the B cell involvement in autoantibody formation, treatment response to rituximab, a B cell-depleting agent, is more favorable with seropositive than seronegative RA [38]. Notably, abatacept, a T cell inhibitor, was also found to associated with a better response among seropositive than seronegative patients in both clinical trials and real-world data [39,40].

## 3. Autoantibodies in Myositis

Myositis represents a systemic autoimmune disease that primarily targets muscles and often involves extra-muscular organs. There are distinct clinical subsets of myositis that can be defined by unique clinical phenotypes and associated autoantibody profiles, suggesting a heterogeneity of disease pathophysiology [41]. To date, various autoantibodies have been identified in myositis, some specific to myositis (myositis-specific autoantibodies, MSAs), and others also found in other autoimmune diseases (myositis-associated autoantibodies, MAAs). A high specificity for myositis and a striking association between individual MSAs and distinct clinical phenotypes make MSAs an excellent tool for subsetting patients and predicting disease courses and outcomes. Despite their usefulness, the new 2017 classification criteria for adult and juvenile myositis by the European League Against Rheumatism/American College of Rheumatology Classification (EULAR/ACR) criteria included only anti-Jo-1 to identify patients with myositis [41]. 

### 3.1. Pathogenic Roles of MSAs

Table 1 summarizes various MSAs in terms of their counterpart autoantigens, prevalence, and associated clinical phenotypes. The most well-known MSAs are anti-aminoacyl-tRNA synthetase (anti-ARS) antibodies directed to the enzymes that catalyze the binding of amino acids to corresponding tRNA to form aminoacyl-tRNA, with anti-Jo-1 or anti-histidyl-tRNA-synthetase being the most common (Table 1).

Many of myositis-specific autoantigens, including ARS, are ubiquitously expressed intracellular proteins involved in various cell functions, particularly transcription or translation (Table 1). They are overexpressed and modified in apoptotic and regenerating muscle fibers [42], some of which are also expressed in tumor cells showing a link between cancer and autoimmunity to muscles [43]. 

While the associations of MSAs with myositis or certain clinical phenotypes of myositis are well-known, only limited information is available about the roles of MSAs in myositis pathogenesis. Animal models and in vitro studies showed that Jo-1 or histidyl-tRNA-synthetase might function as a chemokine and play a role in inducing myositis [44]. A mouse model immunized by Jo-1 further showed inflammation in the muscles and lungs, supporting a pathogenic role of Jo-1 in myositis development [45]. However, it is unknown whether or how anti-Jo-1 antibody regulates chemokine activity of Jo-1. Rather, anti-Jo1 can form IC that induces type I interferon production by plasmacytoid dendritic cells [46].

Recently, a new subset of myositis, referred to as immune-mediated necrotizing myopathy (IMNM), has been identified [47]; this subset of myositis is pathologically characterized by marked muscle fiber necrosis and regeneration with minimal inflammatory infiltrates and serologically by anti-signal recognition particle (anti-SRP) or anti-3-hydroxy-3-methylglutaryl-coenzyme A reductase (anti-HMGCR) antibodies. The latter antibody was particularly positive among IMNM patients with previous statin exposure [48]. While the roles of many other MSAs remain largely unclear, the pathogenic roles of these two antibodies have been better understood by animal models, as well as in vitro studies [49,50,51]. 

Muscle atrophy is one of the typical features of IMNM, together with necrosis and regeneration. Incubation with anti-SRP or anti-HMGCR antibodies led to muscle fiber atrophy and increased expression of atrophy pathway proteins in vitro [49]. In either anti-SRP or anti-HMGCR-positive muscle sections, greater sarcolemmal MAC deposits were observed in lesional areas than in anti-Jo-1-positive muscle tissues, correlating with the degree of muscle necrosis [50]. The pathogenic role of anti-SRP and anti-HMGCR IgG was directly shown in an in vivo study where passive transfer of anti-SRP or anti-HMGCR IgG led to significant muscle weakness, histologic deposits of MAC, and necrotic muscle fibers [51]. 

### 3.2. Assays to Detect MSAs

Historically, MSA identification has been done by protein or RNA immunoprecipitation that compared the molecular weights of precipitated peptides in reference to the positive control. For recently identified novel MSAs, including anti-MDA5, anti-NXP2, and anti-TIF1 antibodies, enzyme linked immunosorbent assays (ELISAs) were developed but their commercial use is still limited due to low prevalence of individual MSAs among patients with myositis. Recently, the line blot assay (LBA; e.g., Euroline Myositis Profile kit by Euroimmun Lübeck, Germany), in which antigens are placed on nitrocellulose as narrow lines, has been introduced particularly to detect multiple autoantibodies simultaneously. Parallel detection of multiple MSAs by the LBA has advantages, considering the low prevalence of each MSA. In a study by Espinosa-Ortega et al. [52], LBA was compared with immunoprecipitation among patients with myositis and showed a good correlation, except for the anti-TIF1γ antibody, for which LBA showed a lower sensitivity than immunoprecipitation. Moreover, LBA detected true positive anti-Jo1 samples that were missed by immunoprecipitation. 

### 3.3. MSAs as a Biomarker 

The most well-established role of MSAs as a biomarker would be based on their highly specific association with myositis and/or distinct clinical subsets (Table 1). Since patient prognosis is different according to individual MSAs and MSA-associated clinical subsets, MSAs are useful, not only for diagnosis, and but also for prognosis prediction and treatment decision of myositis patients. 

#### 3.3.1. Anti-Jo-1 and Other Anti-ARS

Among anti-ARS antibodies, anti-Jo-1 is the most common, found in up to 40% of all myositis patients, while the other anti-ARS antibodies individually constitute less than 5% (Table 1). The most striking feature associated with anti-ARS or anti-Jo-1 is observed as the anti-synthetase syndrome, where interstitial lung disease, myositis, and the presence of anti-ARS are typically co-present with or without other clinical findings, such as the Raynaud phenomenon, arthritis, or mechanic’s hands. Interstitial lung disease (ILD) occurs in up to 90% of individuals possessing anti-Jo-1 antibodies [53]. Since ILD harbors a substantial impact on the survival of patients with myositis, the presence of anti-Jo-1 mandates a careful screening for ILD. 

While patients with individual anti-ARS antibodies share a clinical phenotype called anti-synthetase syndrome, there might be slight differences among patients with different anti-ARS [54,55]. For example, other anti-ARS than anti-Jo-1 antibodies might be associated with ILD in the absence of myositis [55,56]. While other MSAs, such as anti-melanoma differentiation-associated protein 5 (anti-MDA5) antibodies, are also associated with ILD, ILD associated with anti-ARS are often slowly progressive compared to rapidly progressive ILD associated with anti-MDA5 [57]. 

#### 3.3.2. Anti-MDA5

Anti-MDA5 antibodies are DM specific, found in up to 50% of adult DM patients [58]. They can be seen in both classical and clinically amyopathic DM (CADM) but more preferentially in CADM [59,60]. Due to their association with CADM, anti-MDA5 antibodies were also called anti-CADM-140 antibodies before target antigens were identified [59]. Anti-MDA5 antibodies were in fact found to be an independent indicator of rapidly progressive ILD in both classic DM and CADM [60]. Since the prognosis of anti-MDA5-associated ILD is invariably fatal due to respiratory failure, early aggressive treatment has been encouraged [61]. Recently, successful treatment against humoral immunity using rituximab or plasma exchange has been reported against anti-MDA5-positive rapidly progressive ILD among DM patients [61,62]. 

In juvenile DM, anti-MDA5, positive in 7.5% in a UK cohort (n = 285), was associated with amyopathic or hypomyopathic features, mucocutaneous ulcerations, arthritis, and ILD, which, however, was not rapidly progressive [63]. 

#### 3.3.3. Anti-Mi2

Anti-Mi2 is a DM-specific autoantibody characterized by DM rashes (V neck or shawl sign) and good prognosis due to the absence of internal organ involvement [64]. 

#### 3.3.4. Anti-Transcription Intermediary Factor 1 (Anti-TIF1) 

Anti-TIF1 precipitates TIF1γ of 155 Kda with or without TIF1α of 140 KDa, thus formerly called anti-p155 or anti-155/140 antibodies [65]. While they are preferentially associated with both adult and juvenile DM [65,66], their clinical implications are much different according to age at onset of the affected population. 

Anti-TIF1 in adult DM has been well-recognized as an indicator for concurrent cancers [67,68]. Usually, cancers found in patients with active myositis are in advanced stages [69]. A meta-analysis on 312 adult DM patients from six studies suggested that the pooled sensitivity of anti-p155 for diagnosing cancer-associated DM was 78% (95%, CI 45%–94%), specificity was 89% (95%, CI 82%–93%), and the diagnostic OR was 27.26% (95%, CI 6.59%–112.82%) [70]. However, the association between anti-TIF1 antibodies and cancers is unclear in juvenile or young patients, where almost 20%–30% of them are positive for anti-TIF1 [65,66,67].

#### 3.3.5. Anti-Nuclear Matrix Protein 2 (Anti-NXP2)

Anti-NXP2 (also called anti-MJ) antibody is found in adult and juvenile DM [71,72]. In juvenile DM, anti-NXP2 was associated with functional deterioration, such as muscle contractures/atrophy [72], and calcinosis [73,74]. In adult DM, increased risk of cancer [71,75,76,77] and calcinosis [75,76,78] were reported in at least two ethnic groups, while severe muscle involvement, subcutaneous edema, and dysphagia were reported in US cohorts [75,76] in association with the anti-NXP2 antibody.

#### 3.3.6. Anti-Small Ubiquitin-Like Modifier Activating Enzyme (Anti-SAE)

Anti-SAE antibodies are directed to small ubiquitin-like modifier (SUMO) activating enzyme specific for DM. Anti-SAE-positive patients often present as CADM that transitions to overt myositis within months [79]. Although they are linked with a high frequency of systemic features, clinical phenotypes can be different across different ethnicities. For example, frequent dysphagia (30%–78% among anti-SAE-positive patients) has been reported in both UK and Asia [79,80,81], but high prevalence of ILD has only been reported in Asians, with ILD involving two-thirds of anti-SAE-positive patients [80,81].

#### 3.3.7. Anti-SRP and Anti-HMGCR 

These two MSAs are associated with IMNM and poor treatment response [47]. The latter is particularly noted among patients who had statin exposure prior to myositis onset [48]. 

## 4. Autoantibodies in SSc

The pathogenic triad of SSc are: (1) abnormal immune activation as evidenced by disease-specific autoantibodies; (2) obliterative vasculopathy and ischemia resulting from endothelial cell damage/apoptosis, intimal proliferation, luminal obstruction, and vasoconstriction; and (3) tissue fibrosis associated with excess extra-cellular matrix (ECM) accumulation [82]. 

Almost all patients with SSc have autoantibodies [83]. Classic SSc-specific autoantibodies were typically antinuclear antibodies (ANAs), including anti-centromere antibody (ACA), anti-topoisomerase I antibody (ATA), anti-RNA polymerase (RNAP) III antibody, and others [84]. Similar to MSAs, individual SSc-specific ANAs are directed against ubiquitous nuclear proteins and specifically associated with distinctive clinical phenotypes of SSc. Table 2 summarizes various SSc-specific autoantibodies in terms of their counterpart autoantigens, prevalence, and clinical significance. Recognizing their diagnostic and prognostic values, 2013 ACR/EULAR classification criteria for SSc are utilizing anti-centromere antibody, anti-scl70 antibody, and anti-RNAP III antibody to identify patients with SSc [85]. Recent progress on SSc autoantibodies includes that certain SSc autoantibodies are consequences of cross-reactivity between SSc and cancer antigens [86,87,88] and that SSc patients have functional autoantibodies, e.g., anti-receptors antibodies, which may take an important part of SSc pathogenesis [89,90,91,92,93,94,95,96,97,98,99,100,101,102,103,104]. 

### 4.1. Pathogenic Roles of SSc Specific Autoantibodies

While SSc-specific ANAs are associated with SSc or certain clinical phenotypes of SSc, the pathogenic roles of these autoantibodies are largely unknown, particularly considering the nuclear location of the antigens, with limited access by circulating autoantibodies. However, recent studies observed that SSc patients also have non-ANA autoantibodies, particularly those targeting receptors and signaling pathways, with functional consequences (Table 2). In addition, certain SSc-specific autoantibodies suggest that SSc may develop as a consequence of an immune response to the aberrant expression of self-antigens by tumor cells [86].

Although not specific for SSc, anti-endothelial cell antibodies (AECAs) have long been recognized in patients with SSc [89]. A potential pathogenic role of AECA has been suggested by their ability to induce apoptosis of human dermal microvascular endothelial cells in the presence of activated NK cells via the Fas pathway [90]. Recent evidence suggests that AECAs at least include those directed against (intercellular adhesion molecule-1) ICAM-1 [91]. Exposure to IgG from either AECA-positive patients or to anti-ICAM-1 antibodies was similarly found to induce generation of reactive oxygen species (ROS) and vascular cell adhesion molecule-1 (VCAM-1) expression from endothelial cells, indicating a possible contribution of AECAs to SSc vascular lesions [91]. In line with this, AECAs are known to associate with more severe disease phenotypes in terms of internal organ involvement, such as lung fibrosis, pulmonary arterial hypertension (PAH), and ischemic digital lesions [89,92].

In addition to AECA, SSc patients were found to have autoantibodies for renin-angiotensin and endothelin systems, both of which receptors are widely expressed throughout the vascular system and immune cells [93]. Patients with anti-angiotensin II type I receptor (AT1R) and/or endothelin-1 type A receptor (ETAR) antibodies were found to be at high risk for vascular complications, including PAH and ischemic digital lesions [94,95,96]. Increased signaling through the angiotensin and endothelin receptors has been linked to vasculopathy and fibrosis by increasing tissue growth factor β (TGFβ), interleukin-18, and VCAM-1 expression, and ROS generation from human dermal microvascular endothelial cells [96,97]. When healthy C57BL-6 mice were treated with repeated intravenous administrations of IgG from SSc patients positive for anti-AT1R or anti-ETAR antibodies, the lung tissue showed thickened airway vessels and elevated cell density in interstitial tissue [97]. However, the study did not confirm whether these histological changes were specifically induced by anti-AT1R or anti-ETAR antibodies, since it could be due to other agnostic autoantibodies, such as anti-platelet-derived growth factor receptor (anti-PDGFR) antibodies that can be co-present in SSc patient sera.

There are two types of anti-PDGFR among patients with SSc: nonagnostic and agonistic [98]. Only agonistic anti-PDGFR antibodies are SSc-specific, directed for stimulatory epitope of PDGFRα. Agonistic antibodies were found to cause receptor tyrosine phosphorylation of fibroblast PDGFR, which ultimately leads to activation of transcriptional factors to produce extracellular matrix (ECM) and reactive oxygen species (ROS) [99]. Of note, these antibodies stimulated PDGFR for an aberrantly longer time than PDGF itself [99]. In SCID mice, transferred SSc patient IgG or agonistic anti-PDGFR antibodies induced increased collagen deposition and fibroblast activation from the 3D bioengineered skin grafts that contained human keratinocytes and fibroblasts from healthy donor skins [100]. Similar to agonistic anti-PDGFR antibodies, anti-fibrillin-1 antibodies have also been reported to increase ECM via increased TGFβ production [101]. 

Anti-acetylcholine receptor antibodies have been suggested to associate with the gastrointestinal (GI) motility dysfunction of SSc. Certain SSc patients were found to have autoantibodies against myenteric neurons of the GI tract [102], and passive transfer of IgG from SSc patients with high titer anti-myenteric neuronal antibodies led to a decreased myenteric action potential formation [103]. In addition, inhibitory antibodies for type 3 muscarinic receptors (M_3_Rs) were found to have a role in GI dysmotility by working not only on myenteric neurons but also on GI smooth muscles [104]. 

Similarly observed in myositis, certain autoantibodies found in patients with SSc indicates a high risk of concurrent malignancies. For instance, temporal clustering of SSc and cancer onset among patients with anti-RNAP I/III has been noted [105]. Among patients who had cancer diagnoses and produced anti-RNAP I/III antibodies, SSc tends to develop within 2 years of the cancer diagnosis. Interestingly, tumor expression of nucleolar RNAP III was enhanced exclusively in patients with anti-RNAP I/III antibodies than those without. Coupled with later findings that genetic alteration of the POLR3A locus was exclusively found in anti-RNAP III-positive SSc patients with cancer than without and that POLR3A mutation triggered cellular immunity and cross-reactive humoral immune responses [86], the current hypothesis is that malignancy may initiate the scleroderma-specific immune response and drive disease in a subset of scleroderma patients. Anti-RNPC3 antibody directed for the minor spliceosome complex is another autoantibody associated with cancer-associated SSc, particularly among triple negative (negative for ACA, ATA, and anti-RNAP III) SSc patients [87,88]. Anti-RNPC3-positive SSc patients showed a 4-fold increased risk for cancer diagnosis within 2 years of SSc onset, and 89% of them possessed severe ILD at baseline [88].

### 4.2. Assays to Detect SSc Specific Autoantibodies

Very similar to MSAs, SSc-specific ANAs have traditionally been detected by immunoprecipitation, and the recent LBA method is also commercially available for SSc (e.g., Euroline ANA Profile 3 by Euroimmun). When LBA was used in an Australian cohort of 505 patients with SSc, subclassification and disease stratification by autoantibody profile was correlated with characteristic clusters of clinical phenotypes [106], suggesting a usefulness of LBA. 

Functional antibodies have been detected for research purposes and not standardized yet, possibly due to uncertainties of pathogenic epitopes [98,107]. However, due to their association with functional consequences, development of commercial assays is particularly important for these antibodies in terms of prognosis prediction and treatment decision-making on B cell depletion therapy. 

### 4.3. SSc specific Autoantibodies as Biomarkers

#### 4.3.1. Classic SSc Specific ANAs

While ACA, ATA, and anti-RNAP III are the most common SSc-specific ANAs [84], the prevalence of each antibody substantially varies depending on the ethnic groups and detection methods [108]. The presence of SSc-specific ANAs among patients with Raynaud’s phenomenon indicates a high risk for future SSc [109].

##### ACA

ACA, first described as an indicator of CREST (Calcinosis, Raynaud’s phenomenon, Esophageal hypomotility, Sclerodactyly, and Telangiectasia) syndrome, is now considered as a variant of limited cutaneous SSc [110]. According to the European Scleroderma Trials and Research group (EUSTAR) database, ACA is the most common autoantibody for limited cutaneous SSc found in up to 50% of them [84]. Typically, ACA-positive SSc patients tend to have limited cutaneous SSc and better prognosis due to a lack of internal organ involvement, such as ILD, cardiomyopathy, or renal crisis [111], except for PAH, which can occur later in the course of the limited disease [112]. In this regard, the evidence-based DETECT algorithm for the identification of SSc patients at high risk for developing PAH includes ACA as one of the useful screening items [113]. 

##### ATA

Detected in 20%–30% of SSc patients, ATAs are typically associated with diffuse cutaneous SSc and internal organ involvement [111]. Approximately two-thirds of ATA-positive SSc patients were found to have diffuse cutaneous SSc, and ILD occurs in 70% of them [111]. Since the main cause of death among SSc is ILD and PAH, ATA is a marker of poor prognosis [114]. Other important organ involvements associated with ATA include cardiomyopathy and peripheral vascular complications such as digital ulcer (DU) and gangrene, particularly early in the disease course [115]. 

##### Anti-RNAP III

The frequency of Anti-RNAP III in SSc patients varies among ethnic groups: higher frequency in North American Caucasian and UK patients (~25%) in comparison with French (~9%) or Japanese patients (~5%), with a pooled prevalence of 11% [116]. Majority of anti-RNAP III-positive patients show diffuse cutaneous SSc with rapid progression followed by regression of skin thickening [116,117]. Typically, anti-RNAP III positivity has been associated with a higher risk of renal crisis, but severe ILD was rare [115,118]. Another clinical phenotype associated with anti-RNAP III is gastric antral vascular ectasia (GAVE), observed in the EUSTAR database where 48% of patients with GAVE were anti-RNAP III-positive, compared with 16% of those without [119], suggesting anti-RNAP III involvement in SSc-related vasculopathy. While a recent study showed that anti-RNAP III antibody response is triggered by neoantigens by cancer-related genetic alteration, the prevalence of cancers among Anti-RNAP III-positive patients is generally low [86].

##### Anti-U3 RNP (Anti-Fibrillarin) 

Anti-U3 RNP recognizes a 34kD component of the U3 RNP complex called fibrillarin. They are detected in −10% of SSc patients, with a predilection for male sex and African-American patients [115,120,121]. Anti-U3 RNP has been suggested as a marker for severe SSc, reportedly linking with diffuse cutaneous SSc and diverse internal organ derangements of cardiac, renal, musculoskeletal, pulmonary, and gut systems [115,120,122,123], particularly in the form of pulmonary arterial hypertension; thus, poor survival [120]. 

##### Anti-Th/To

Anti-Th/To antibodies primarily bind to two RNA processing enzymes, RNase P and RNase MRP. They are relatively specific for SSc, seen in 1%–10% among patients with SSc. In general, they are associated with limited cutaneous SSc but have a high frequency of internal organ involvements, such as ILD, PAH, pericarditis, and/or renal crisis [115,124,125]; hence, a poor prognosis [124]. 

##### Anti-U11/U12 RNP

Reactive to both U11 and U12 RNA, anti-U11/U12 RNP antibody is approximately found in 3% of US SSc patients (15/462) and associated with severe ILD and high mortality [126].

##### Anti-Eukaryotic Initiation Factor 2B

Observed in approximately 1% of British patients with SSc (7/548), anti-eukaryotic initiation factor 2B antibody is a cytoplasmic antibody strongly associated with diffuse cutaneous SSc and ILD [127]. 

#### 4.3.2. Functional Antibodies

As mentioned above, functional antibodies are typically directed for receptors and signaling pathways involved in fibrosis, the vascular system, and neurotransmission. The exact prevalence of these antibodies is not known due to the unavailability of commercial assays, but the successful treatment of SSc skin and lung using rituximab supports the clinical importance of functional autoantibodies of SSc [128,129,130]. 

## 5. Autoantibodies in ANCA-Associated Vasculitis

Antineutrophil cytoplasmic antibody (ANCA)-associated vasculitides (AAV) are a spectrum of systemic necrotizing vasculitis predominantly affecting small vessels of diverse organs. They are subdivided into three distinctive disease entities of granulomatosis with polyangiitis (GPA), microscopic polyangiitis (MPA), and eosinophilic GPA (EGPA), which are associated with ANCA in more than 90% of cases under current clinical assays [131]. GPA and EGPA share extravascular necrotizing granulomatous inflammation or granulomatosis that most often affects respiratory tracts but potentially any organs [132]. While autoantibodies or ANCAs are directly involved in the pathogenesis of the disease [133,134], AAV is featured by a lack of immune deposits (pauci-immune).

Antineutrophil cytoplasmic antibodies (ANCAs) are autoantibodies directed against diverse antigens within the primary granules of neutrophils and the lysosomes of monocytes [132]. ANCAs may develop against any of the granular proteins, but the clinically relevant ones are those directed against myeloperoxidase (MPO) and proteinase 3 (PR3). While both MPO and PR3 are cytoplasmic antigens, ethanol fixation leads to dissolution of proteins within the neutrophil primary granules, leaving a perinuclear pattern for MPO-ANCA and a cytoplasmic pattern for PR3-ANCA [135]. The distinction between different AAVs based on ANCA patterns is far from perfect; PR3-ANCAs were present in about two-thirds of patients with GPA but also in one-quarter of patients with MPA, whereas MPO-ANCAs were in the majority of patients with MPA but also in up to one-quarter of patients with GPA [136]. Since the large-scale genome-wide association study demonstrated that the strongest genetic association comes with ANCA antigen specificity, rather than with a specific clinical syndrome [137], differentiating between patients with MPO-ANCAs or PR3-ANCAs might be more relevant, instead of distinguishing between patients with GPA, MPA, and EGPA [136]. 

Although current classification systems do not include ANCA as a diagnostic tool for AAV, the 2010 EULAR have pointed to considering ANCA in diagnostic and classification criteria for systemic vasculitis [138], and the 2012 revised Chapel Hill Consensus Conference Nomenclature of Vasculitides (CHCC 2012) called for adding a prefix of ANCA specificity to each AAV (i.e., MPO-ANCA GPA, and ANCA-negative EPA) [135], acknowledging the clinical relevance of ANCA in AAV. 

### 5.1. Pathogenic Roles of ANCA

Direct evidence for ANCA pathogenicity comes from an animal model in which the passive transfer of MPO-ANCA resulted in pauci-immune crescentic GN and necrotizing vasculitis in small vessels [133]. Effective disease prevention by neutrophil depletion in this model further indicates a central role of neutrophils in ANCA-induced crescentic GN [139]. Unlike an animal model by MPO-ANCA, the development of PR3-ANCA-induced animal models has been less successful in earlier studies, possibly due to lack of PR3 expression on murine neutrophils and different FcR affinities for IgG across species in mice/rodents [140]. However, pathogenic PR3-ANCA has been also demonstrated in mice with a humanized immune system [134].

Figure 1 summarizes the pathogenic processes elicited by ANCAs. While MPO and PR3 are intracellular antigens, they can be accessed by antigen-presenting cells under a specific condition called priming [141]. Neutrophils primed with inflammatory stimuli (cytokines, lipopolysaccharides, complement fragments, etc.) release PR3 or MPO from primary granules to the cell surface [142], which permits ANCAs to interact with them. Cell surface displays of ANCA antigens can be achieved by their binding to the cell surface after release from primed or activated neutrophils [143] or by direct translocation of these antigens towards cellular membrane under inflammatory microenvironments [144]. Importantly, surface displays of ANCA antigens can be induced on apoptotic neutrophils independent of priming [145,146], which provides an alternative mechanism beyond neutrophil activation to allow interaction between ANCAs and ANCA antigens. 

ANCA acts via antigen crosslinking by ANCA Fab or via FcγR engagement [147,148]. ANCA-activated neutrophils generate a respiratory burst with a production of oxygen radicals, degranulate destructive enzymes, and extrusion of NETs [149]. NETosis contributes to AAV by causing endothelial damage, lymphocyte stimulation, and activation of alternative complement pathways [150]. Since NETs further expose ANCA antigens to ANCA, the vicious cycle forms to perpetuate inflammation [151,152]. Besides neutrophil activation, ANCA also promotes neutrophil adherence on capillary endothelium by neutrophil integrin activation and upregulation [153]. Vascular injury occurs via ANCA-activated neutrophils adherent to endothelial cells [149] or via direct effects of internalized ANCA antigens by endothelial cells [154]. When internalized, PR3 induces endothelial apoptosis, whereas MPO induces intracellular oxidants [154]. While PR3 ANCA or MPO ANCA utilize a number of similar mechanisms to cause disease [132], different clinical phenotypes in association with ANCA specificity indicates there could be different inflammatory pathways for two different ANCAs [154,155]. However, it is largely unknown if and how PR3 and MPO ANCAs behave differently. 

### 5.2. Assays to Detect ANCAs

Laboratories have used indirect immunofluorescence (IIF) assays as a screening to detect ANCA, semi-quantitatively reporting serum dilution factors. However, the positive result should be confirmed by antigen-specific assays. Moreover, atypical ANCAs that do not target PR3 or MPO (positive IF but negative ELISA) can be found in a range of nonvasculitic conditions, such as inflammation (drugs, chronic infection, cancer, and other autoimmune diseases). Since it is technically difficult to differentiate perinuclear ANCA (or MPO ANCA) patterns from antinuclear antibodies (ANA) based on ethanol fixation, some laboratories use both formalin and ethanol fixation to distinguish ANCAs from ANAs [156]. 

The first-generation ANCA ELISA developed in the late 1990s was a direct method immobilizing ANCA antigens to the surface of an ELISA plate, which often has a lower sensitivity than IIF, despite its high specificity (>95%) [157]. The second-generation capture ELISA uses secondary antibodies attached to the plate that specifically capture ANCA antigens, showing a better sensitivity than the direct ELISA [158,159]. The third-generation anchor ELISA immobilizes antigens to the ELISA plate via a bridging molecule to preserve epitope binding sites that could have been masked by capturing antibodies. The anchor ELISA was better than the direct ELISA in terms of sensitivity [160,161], which, in some studies, showed a superior sensitivity to capture ELSIA and IIF [162]. 

While the 1999 international consensus statement recommends a testing strategy that adopts IIF as an initial screening tool followed by confirmation with ELISA [163], the updated 2017 international consensus statement of ANA testing proposes that high-quality immunoassays can be used as the primary screening method for patients suspected of having the AAV without the categorical need for IIF, acknowledging the excellent sensitivity of the recent capture and/or anchor assays [136]. However, approximately 10% of patients with clinical features and pathology of AAV are negative for ANCA IIF [156]. 

### 5.3. ANCA as Biomarkers

As mentioned above, the positivity of MPO or PR3 ANCA in AAV ranges above 90% in generalized GPA and MPA, particularly in their active phase [131], and thus, may improve diagnostic sensitivity when used under high suspicion of vasculitis. In particular, rapid ANCA tests, when urgent therapeutic decisions are required, might be helpful as an adjunct before biopsy results are pending or not available.

In addition to diagnostic benefits, studies have shown that ANCA specificity is the major determinant of clinical manifestations and outcomes. In the analysis on 502 ANCA-positive patients with biopsy-proven AAV, MPO-ANCAs were present in >80% of patients with isolated crescentic glomerulonephritis, whereas PR3-ANCAs were in >80% of patients with lung cavities or upper airway involvement [164]. Similarly, the RAVE trial (rituximab versus cyclophosphamide for ANCA-associated vasculitis), the largest randomized controlled trial in AAV so far, has shown a higher rate of renal involvement (79% versus 59%) and peripheral neuropathy (27% versus 15%) among patients with MPO-ANCAs than with PR3-ANCAs and the opposite pattern with constitutional symptoms (53% versus 71%), ear/nose/throat involvement (33% versus 70%), and lung nodules or cavities (12% versus 28%) [165,166]. According to a study on 673 patients with GPA or MPA from five different randomized controlled trials, it was found that the addition of ANCA specificity to a clinical-only clustering strategy resulted in a better separation of patients with different clinical phenotypes and outcomes [167]. The study showed a higher relapse with PR3-ANCA but higher mortality without PR3-ANCA, indicating different treatment outcomes and relapse rates according to the ANCA specificity. A consistent finding was observed by a systematic review in which patient survival was worse in MPO-AAV [168]. However, outcomes on renal survival are controversial, with some studies reporting a better prognosis with MPO-ANCA [169] and others with PR3-ANCA [170,171].

In the RAVE trial [165] that compared cyclophosphamide and rituximab against AAV, ANCA specificity was not predictive of complete remission at six months in either the entire cohort or in each of the two treatment groups. However, a post hoc analysis showed that complete remission at six months was achieved with a similar frequency in both treatment groups among 66 MPO-ANCA-positive patients (61% versus 64%), whereas rituximab was significantly more effective than cyclophosphamide among 131 PR3-ANCA-positive patients (65% versus 48%; *p* = 0.04) [166]. 

According to the WEGENT trial that compared azathioprine versus methotrexate for maintenance therapy of AAV, only 30% of patients with GPA or MPA remained relapse-free during approximately 12 years of follow-up [172]. The relapse of AAV was often preceded by rising titers of ANCA, and there was rarely a relapse among patients with undetectable ANCAs [173]. The predictability of the ANCA titer change was particularly relevant for renal than nonrenal relapse, regardless of ANCA specificity [174]. While certain patients remain in clinical remission despite high titers of ANCA, the epitopes recognized by ANCAs were found to be different between inactive versus active phases, suggesting that ANCAs reacting with pathogenic epitopes are more relevant than titers [175]. 

## 6. Conclusions

The presence of disease-specific or disease-associated antibodies is the hallmark of the autoimmunity of the given rheumatic disease. Each rheumatic disease exhibits a unique pattern of antibody profiles that provides useful information for diagnosis, clinical phenotypes, and prognosis. Since the antibody titers are often correlated with disease activity or relapse, it is expected that they are actively involved in disease pathogenesis. Although exact pathophysiologic roles of the disease-specific or disease-associated antibodies remain largely unknown, tremendous progress has been made on this issue during the past decade. RA, myositis, SSc, and AAV are the prototype diseases in which disease-specific, or sometimes, disease-associated, autoantibodies are not only acknowledged as biomarkers to help diagnosis, predict prognosis, and guide treatment decision-making, but also play important roles to perpetuate and aggravate the disease process. Therefore, values of these autoantibodies as a biomarker would be potentiated by understanding their pathogenic roles closely linked with clinical observations. 

## Figures and Tables

**Figure 1 ijms-21-01382-f001:**
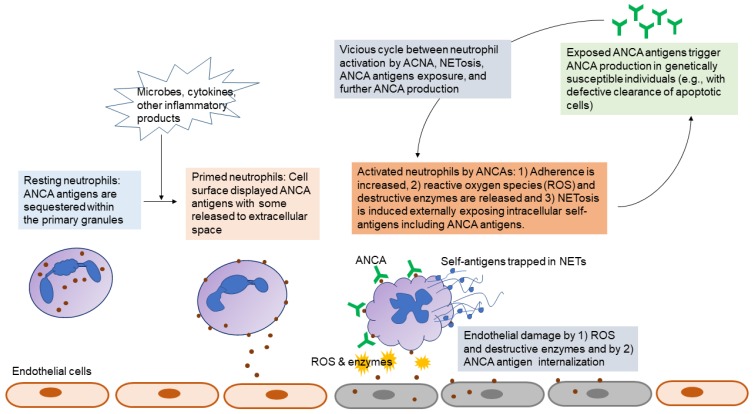
Pathogenic roles of antineutrophil cytoplasmic antibodies (ANCAs).

**Table 1 ijms-21-01382-t001:** Summary of myositis-specific autoantibodies.

MSAs	Autoantigens	Representative Physiologic Functions of Autoantigens	Prevalence in Adults	Associated Clinical Phenotypes
**Traditional Autoantibodies**				
Anti-ARS	Aminoacyl-tRNA-synthetase	Translation	20%–40%	Anti-synthetase syndrome
anti-Jo-1	histidyl-tRNA-synthetase		15%–20%	
anti-PL-7	threonyl-tRNA-synthetase		<5%	
anti-PL-12	alanyl-tRNA-synthetase		<5%	
anti-OJ	isoleucyl-tRNA-synthetase		<5%	
anti-EJ	glycyl-tRNA-synthetase		<5%	
anti-KS	asparaginyl-tRNA-synthetase		<5%	
anti-Ha	tyrosyl-tRNA-synthetase		<5%	
anti-Zo	phenylalanyl-tRNA-synthetase		<5%	
Anti-SRP	Signal recognition particle	Translocation of newly synthesized proteins across endoplasmic reticulum	<10%, but some studies reporting 20% among myositis patients	Immune-mediated necrotizing myopathy or cardiac involvement
Anti-Mi2	Nucleosome remodeling deacetylase	Chromatin remodeling	<10%	Rashes of DM
**Recent or Newly Found Antibodies**				
Anti-MDA5	Melanoma differentiation-associated protein 5	Pattern recognition receptor for viral RNA	20%–50% in Asians 7%–13% in Caucasians	Rapidly progressive ILD in classic DM and CADM
Anti-Mi2	Nucleosome remodeling deacetylase	Chromatin remodeling	<10%	Rashes of DM
Anti-TIF1	Transcriptional intermediary factor 1	Chromatin remodeling; ubiquitination	10%–20% of adult DM (but higher in cancer-associated myositis)	Malignancy in adult DM
Anti-NXP2	Nuclear matrix protein 2	Chromatin remodeling; activation of p53	<10%	Malignancy in adult DM; calcinosis and severe myositis in juvenile DM
Anti-HMGCR	3-hydroxy-3-methylglutaryl-coenzyme A reductase	Cholesterol synthesis	<10%	Immune-mediated necrotizing myopathy; previous statin exposure
Anti-SAE	SUMO activating enzyme subunit 1	Post-translational protein modification	<5%	Transition from CADM to overt myositis; dysphagia in both Caucasians and Asians but ILD only in Asians.

ARS: aminoacyl-tRNA synthetase, CADM: clinically amyopathic dermatomyositis, DM: dermatomyositis, HMGCR: 3-hydroxy-3-methylglutaryl-coenzyme A reductase, ILD: interstitial lung disease, MDA5: melanoma differentiation associated protein 5, NXP2: nuclear matrix protein 2, RNA: ribonucleic acid, tRNA: transcriptional RNA, SRP: signal recognition particle, SUMO: small ubiquitin-like modifier, TIF1: transcriptional intermediary factor 1, and MSAs: myositis-specific autoantibodies.

**Table 2 ijms-21-01382-t002:** Summary of scleroderma-related autoantibodies.

Autoantibody	Autoantigens	Representative Physiologic Functions of Autoantigens	Prevalence	Associated Clinical Phenotypes
**Traditional SSc-Specific ANAs**				
Anti-centromere	Centromeric protein	Contains histone H3 and involves epigenetic process	10%–50%	Limited cutaneous SSc PAH CREST DU in late phase
Anti-topoisomerase I	Topoisomerase I	Enzyme that cuts, relaxes, and reanneals one of the two strands of double-stranded DNA.	20%–30%	Diffuse cutaneous SSc ILD DU in early phase
Anti-RNA polymerase	RNA polymerase	Transcription	5%–30%	Diffuse cutaneous SSc Rapid progression of skin-thickening, renal crisis, GAVE, and cancer
**Novel SSc-Specific ANAs**				
Anti-U3 RNP	Fibrillarin complexed with small nucleolar RNA U3	Pre-rRNA processing localized in the fibrillar region of the nucleolus.	<10%	Both limited and diffuse cutaneous SSc ILD, PAH, renal crisis, and lower GI involvement in early phase
Anti-Th/To	Human RNase MRP complex	Mitochondrial RNA processing; Pre-rRNA processing	<10%	Limited cutaneous SSc ILD PAH
Anti-U11/U12 RNP	Small nucleolar RNA U11/U12	Noncoding RNA in the minor spliceosome complex that activates the alternative splicing	<5%	Both limited and diffuse cutaneous SSc ILD
**Novel Functional SSc Antibodies**				
Anti-PDGFR	Platelet-derived growth factor receptor	PDGF receptor on fibroblasts	NA	Vasculopathy and ILD
Anti-M_3_R	Type 3 muscarinic receptor	Acetylcholine receptor on myenteric neurons and visceral smooth muscles	NA	GI dysmotility
Anti-ICAM-1 or AECA	ICAM-1 or endothelial cells	Adhesion molecules on endothelial cells	NA	Vasculopathy
Anti-AT1R	Angiotensin II type 1 receptor	Receptor for angiotensin on visceral smooth muscles	NA	Vasculopathy, ILD
Anti-ETAR	Endothelin-1 type A receptor	Receptor for endothelin A on visceral smooth muscles	NA	Vasculopathy, ILD

AECA: anti-endothelial cell antibody; ANA: antinuclear antibody; AT1R: angiotensin II type 1 receptor; CREST: Calcinosis, Raynaud’s phenomenon, Esophageal hypomotility, Sclerodactyly, and Telangiectasia; DU: digital ulcer; ETAR: endothelin-1 type A receptor; GAVE: gastric antral vascular ectasia; GI: gastrointestinal; ICAM-1: intercellular adhesion molecule-1; ILD: interstitial lung disease; M_3_R: type 3 muscarinic receptor; NA: nonapplicable; RNA: ribonucleic acid; RNP: ribonucleoprotein; PAH: pulmonary arterial hypertension; PDGFR: platelet-derived growth factor receptor; and SSc: scleroderma.

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
