# Peer review of "Autoantibody Biomarkers in Rheumatic Diseases"

_ijms, 2020, doi:10.3390/ijms21041382_

Round 1

Reviewer 1 Report

This review is well-wrriten, easy to understand  the current knowledge regarding autoantibodies. One point which I would like to suggest is that the authors might want to demonstrate the pathophysiology related to ANCA in AAV using the figure.  

Author Response

We thank the reviewer. As suggested, we added a Figure to demonstrate pathogenic processes related to ANCA in AAV.

Reviewer 2 Report

The authors present a thorough Review, covering autoantibodies in RA, myositits, scleroderma and AAV. There are only minor issues to be adressed. The manuscript would benefit from an English style and spell check. ANCA is for example spelled ACNA several times. There are some minor corrections/re formulation issues.

Author Response

Reviewer #2

The authors present a thorough Review, covering autoantibodies in RA, myositits, scleroderma and AAV. There are only minor issues to be adressed. The manuscript would benefit from an English style and spell check. ANCA is for example spelled ACNA several times. There are some minor corrections/re formulation issues.

[Response]

We thank the reviewer for this comment, and corrected typos and grammars. 

Reviewer 3 Report

Overall informative and well-written. 

In order to improve readability, some amendments can be required

-taking into account the main topic of the review "antibodies as biomarkers", the sections dedicated to pathogenic aspects should be reduced or omitted. 

-a section focusing on challenges and opportunities of different autoantibodies testing methods should be incorporated (e.g.,  IIF Vs. solid assays Vs Multi-Analyte Diagnostic Platform)

-Figure 1 can be omitted. 

-The tables can be restructured dividing biomarkers between traditional and novel/upcoming ones for each condition. 

Author Response

We thank the reviewer for thorough comments.

Since we believe that the knowledge on pathogenic roles of biomarkers is essential to better understand the biological and clinical relevance of biomarkers, we only reduced the pathogenesis sections.

We added a section on assays for each disease as follows.

1.3. Assays to detect ACPAs

In most of cases, ACPAs are usually measured by anti-cyclic citrullinated peptide (anti-CCP) assays. Based on the finding that anti-perinuclear factor highly specific for RA is indeed autoantibody directed to citrullinated filaggrin, the first-generation anti-CCP (anti-CCP1) utilizes filaggrin-derived cyclic than linear citrullinated peptides to improve antibody recognition [34]. Unlike anti-CCP1, the second and third generation tests (anti-CCP2 and anti-CCP3) are no longer based on filaggrin derived peptides but use mixture of synthetic cyclic peptides selected from libraries of citrullinated peptides of synovial self-antigens, which mimic conformational epitopes [34]. It is well established that the anti-CCP2 possesses a superior sensitivity (around 70-80%) than anti-CCP1 while maintaining very high specificity (98–99%) [35]. Until now, anti-CCP2 is considered as the gold standard of testing for ACPA [36]. In recent years, new ACPA tests such as anti-CCP3 assay, anti-mutated citrullinated vimentin, or others have been commercially introduced. However, they have not shown a clear advantage over anti-CCP2 assay [34].

2.2. Assays to detect MSAs

Historically, MSA identification has been done by protein or RNA immunoprecipitation that compared the molecular weights of precipitated peptides in reference to the positive control. For recently identified novel MSAs including anti-MDA5, anti-NXP2, anti-TIF1 antibodies, enzyme linked immunosorbent assay (ELISA) were developed but their commercial use is still limited due to low prevalence of individual MSAs among patients with myositis. Recently, the line blot assay (LBA; e.g., Euroline Myositis Profile kit by Euroimmun Lübeck, Germany), in which antigens are placed on nitrocellulose as narrow lines, has been introduced particularly to detect multiple autoantibodies simultaneously. Parallel detection of multiple MSAs by the LBA has advantages considering low prevalence of each MSA. In a study by Espinosa-Ortega et al [52], LBA was compared with immunoprecipitation among patients with myositis and showed a good correlation except for anti-TIF1γ antibody for which LBA showed a lower sensitivity than immunoprecipitation. Moreover, LBA detected true positive anti-Jo1 samples that were missed by immunoprecipitation.  

3.2. Assays to detect SSc specific autoantibodies

Very similar to MSAs, SSc specific ANAs have traditionally been detected by immunoprecipitation, and recent LBA method is also commercially available for SSc (e.g., Euroline ANA Profile 3 by Euroimmun). When LBA was used in an Australian cohort of 505 patients with SSc, subclassification and disease stratification by autoantibody profile was correlated with characteristic clusters of clinical phenotypes [106], suggesting a usefulness of LBA.

Functional antibodies have been detected for research purposes and not standardized yet possibly due to uncertainties of pathogenic epitopes [98, 107]. However, due to their association with functional consequences, development of commercial assays is particularly important for these antibodies in terms of prognosis prediction and treatment decision making on B cell depletion therapy.

4.2. Assays to detect ANCAs

Laboratories have used indirect immunofluorescence (IIF) assays as a screening to detect ANCA, semi-quantitatively reporting serum dilution factors. However, the positive result should be confirmed by antigen specific assays. Moreover, atypical ANCAs that do not target PR3 nor MPO (positive IF but negative ELISA) can be found in a range of non-vasculitic conditions such as inflammation. (drugs, chronic infection, cancer, other autoimmune diseases). Because it is technically difficult to differentiate perinuclear ANCA (or MPO ANCA) patterns from antinuclear antibodies (ANA) based on ethanol fixation, some laboratories use both formalin and ethanol fixation to distinguish ANCAs from ANAs [156].  

The first-generation ANCA ELISA developed in late 1990s was a direct method immobilizing ANCA antigens to the surface of ELISA plate, which often has a lower sensitivity than IIF despite high specificity (>95%) [157]. The second-generation capture ELISA uses secondary antibodies attached to the plate that specifically capture ANCA antigens, showing a better sensitivity than the direct ELISA [158, 159]. The third-generation anchor ELISA, immobilizes antigens to the ELISA plate via a bridging molecule to preserve epitope binding sites that could have been masked by capturing antibodies. The anchor ELISA was better than the direct ELISA in terms of sensitivity [160, 161], which, in some studies, showed a superior sensitivity to capture ELSIA and IIF [162].

While the 1999 international consensus statement recommends a testing strategy that adopts IIF as an initial screening tool followed by confirmation with ELISA [163], the updated 2017 international consensus statement of ANA testing proposes that high-quality immunoassays can be used as the primary screening method for patients suspected of having the AAV without the categorical need for IIF, acknowledging the excellent sensitivity of the recent capture and/or anchor assays [136]. However, approximately 10% of patients with clinical features and pathology of AAV are negative for ANCA IIF [156].

Reviewer 4 Report

This article reviews serological findings in RA, myositis, SSc and vasculitis.   While a selective focus is fine, some important specificities and mechanisms are not considered.  

An important issue concerns antibody assays. While the discussion on RA considers ACPAs, many of the studies cited are based on anti-CCP assays. I think that the authors should address the relationship between ACPAs and anti-CCP and delineate more completely the protein targets of ACPA.

While the pathogenicity of ACPAs has been uncertain and to an extent controversial, it is worth citing information on the role of ACPAs in pain. 

Some consideration of assays would also be useful for ANCAs.

The article is detailed and long.  Additional graphics would be helpful  

Author Response

Reviewer #4

This article reviews serological findings in RA, myositis, SSc and vasculitis. While a selective focus is fine, some important specificities and mechanisms are not considered.  

An important issue concerns antibody assays. While the discussion on RA considers ACPAs, many of the studies cited are based on anti-CCP assays. I think that the authors should address the relationship between ACPAs and anti-CCP and delineate more completely the protein targets of ACPA.

While the pathogenicity of ACPAs has been uncertain and to an extent controversial, it is worth citing information on the role of ACPAs in pain.

Some consideration of assays would also be useful for ANCAs.

The article is detailed and long. Additional graphics would be helpful.

[Response]

As suggested by the reviewer, we added sections to address assays for individual diseases (please refer to our answers to reviewer #3) and added a comment citing a reference for a role of ACPA in pain as follows.

Pain induction

While pain may not be not a causal factor for RA, it often precedes clinical or ultrasonic synovitis. According to the study by Wigerblad et al., IL-8, released from osteoclasts in an autoantibody-dependent manner, produces pain by activating sensory neurons independently of inflammation, which may explain the disconnect between pain and inflammation in certain settings of RA [34].  

The original Figure 1 has been omitted by a suggestion of reviewer #3 but updated Figure 1 on the role of ANCAs in AAV has been newly added (please refer to our answer to reviewer #1).

Round 2

Reviewer 3 Report

The manuscript has been improved and main comments addressed